🔓 | **Open Peer Review** | Clinical Microbiology | Research Article

# Diabetes affects the composition of the respiratory tract microbiome and transcriptome in patients with viral pneumonia

Changrui Huang,[1,2] Qinqi Feng,[3] Bang Yu,[1,2] Hao Zou,[1,2] Yashi Cai,[1,2] Jian Liu,[2] Demin Li,[2] Hongchun Zhang,[2] Xiaohui Zou[4]

**ABSTRACT**  Research shows that patients with viral pneumonia complicated by diabetes have a worse prognosis and higher mortality. Our study aimed to assess the effect of diabetes on respiratory tract microbes and the transcriptome in patients with viral pneumonia. We included 76 subjects from China-Japan Friendship Hospital, including 16 healthy people, 17 patients with viral pneumonia and diabetes (VD), and 43 patients with viral pneumonia without diabetes (VP). We collected their sputum samples for both metagenomic and 16S rRNA sequencing and collected blood samples for RNA sequencing. In transcriptome analysis, the VD group downregulated the expression of PTCH1 and upregulated the expression of ANK1, RBM38, BPGM, CRYM, TAL1, and HBD. The differential pathways are mainly reflected in the formation, development, and maintenance of red blood cells, the activity of immunoglobulins, and the membrane transport and transportation of substances. There is a significant difference in microbial diversity between the two groups. Both analysis methods demonstrate a significant increase in the abundance of *g__Treponema*, *s__Treponema_denticola,* and *s__Campylobacter_rectus* in the VP group. The host genes AGAP1, RNF182, and ANKRD9 are particularly closely associated with microorganisms. Our results suggest that diabetes may inhibit the expression of genes related to immune regulation, energy metabolism, and oxygen utilization in patients with viral pneumonia. Meanwhile, we predict that VD may be associated with a decrease in microbial diversity and a decline in microbial functions in cellular processes, environmental adaptation, metabolism, and genetic activity. These abnormalities can worsen the course of viral pneumonia and affect the prognosis of patients.

**IMPORTANCE**  We used 16S rRNA and metagenome sequencing to analyze the respiratory microbial composition of patients with viral pneumonia complicated by diabetes (VD) and patients with viral pneumonia without diabetes (VP) and used transcriptome sequencing to compare the gene expression of patients in VD, VP, and healthy people. Our results indicate significant differences in gene expression and respiratory microbiota profiles between VD and VP. VD may inhibit the immune regulatory response and affect cell energy metabolism and oxygen transport and utilization by regulating related gene pathways. The abundance of *Treponema denticola* in the VP group was significantly higher than that in the VD group. We predicted that the functions of differential microorganisms may be related to cellular processes, environmental information processing, genetic information processing, human diseases, and metabolism. This study found characteristic biomarkers related to viral pneumonia with diabetes, providing a new strategy for further research and clinical treatment.

**KEYWORDS**  microbiota, viral pneumonia with diabetes, transcriptome, traditional Chinese medicine

**Peer Reviewers** Mingkun Li, Beijing institute of genomics, Chinese Academy of Sciences, Beijing, China; Hui Wen, George Washington University, Milken Institute School of Public Health, Washington, DC, USA

Address correspondence to Hongchun Zhang, 13701226664@139.com, or Xiaohui Zou, zouxiaohui17@163.com.

Changrui Huang and Qinqi Feng contributed equally to this article. Author order was determined by drawing straws.

The authors declare no conflict of interest.

See the funding table on p. 13.

Viral pneumonia is a kind of acute respiratory infectious disease, which has the characteristics of a wide range of infections, strong infectivity, and sudden onset (1, 2). The body may experience fever, cough, sore throat, and even shock and respiratory failure in severe cases (3). At present, most viral pneumonia cases do not have specific antiviral drugs; you can only take symptomatic treatment. Recent studies have suggested that COVID-19 patients with diabetes are more severe, with twice the mortality rate of non-diabetic patients and a worse prognosis (4, 5). The respiratory microbiota of people with diabetes also differs from that of healthy people, with the high-glycemic environment providing favorable growth conditions for certain disease-causing microorganisms, especially bacteria and fungi (6–9). However, the mechanism by which these changes occur in patients with diabetes and viral pneumonia remains unclear. In traditional Chinese medicine, diabetes was first noted as "Xiao ke" in the "Yellow Emperor's Inner Canon." The core pathogenesis is yin deficiency with dry heat (10). And the basic principle of traditional Chinese medicine in treating diabetes is heat-clearing and yin-nourishing. A series of classic prescriptions has achieved significant therapeutic effects.

In the field of viral pneumonia research, transcriptomics technology can deeply analyze the gene expression changes of host cells after viral infection, reveal the pathogenesis of viral pneumonia at the molecular level, and provide key clues for exploring potential biomarkers and therapeutic targets. In addition to viruses, microbial communities such as bacteria and fungi in the respiratory tract have also attracted much attention. The differences in the composition of these microorganisms in healthy and diseased states, their interactions, and their effects on the course of viral infection are gradually being revealed. From the perspective of transcriptomics and microbiomes, this study collected and analyzed sputum and blood samples from patients with viral pneumonia complicated by diabetes and patients without viral pneumonia complicated by diabetes, discussed how diabetes affects the course and prognosis of patients with viral pneumonia, and sought the correlation between the host transcriptome and respiratory microbiome. This can also be used to explain the modern biological mechanism of "yin deficiency with dry heat" in "Xiaoke" in traditional Chinese medicine to a certain extent.

## MATERIALS AND METHODS

### Study population

Patients with viral pneumonia included in the study must meet the following diagnostic criteria: (1) clinical symptoms: cough, sputum, sore throat, runny nose, chest pain, and other respiratory symptoms, as well as fever, headache, generalized aches and pains, tiredness, and other systemic symptoms; (2) CT imaging changes: lobular distribution of hairy glass shadows, small nodular foci, reticulation cords and stripes, bronchioles, vascular bundles thickening, lobes, segments of the actual change shadow; may be accompanied by a small amount of unilateral or bilateral pleural effusion or mediastinal lymph nodes; (3) any respiratory sample testing positive for viral pathogens (including sputum, bronchial aspirate, bronchoalveolar lavage fluid, etc.). The following circumstances will not be included if they exist: immunosuppression; poorly compliant mental disorders; coexisting hematological diseases; coexisting malignant tumors; coexisting multiple organ failure; coexisting acute respiratory distress syndrome; coexisting active tuberculosis; pregnant and lactating women; and other populations not eligible to participate in this study.

The inclusion criteria for patients with viral pneumonia complicated by diabetes are as follows: based on the above viral pneumonia diagnostic criteria, if the patient has a clear history of diabetes in the past or is newly diagnosed with diabetes during hospitalization, they will be included. The criteria for newly diagnosed diabetes are the patient has typical symptoms such as polydipsia, polyphagia, polyuria, weight loss,

and fasting blood glucose ≥7.0 mmol/L, or OGTT 2 h blood glucose ≥11.1 mmol/L, or glycosylated hemoglobin ≥6.5%, or random blood glucose ≥11.1 mmol/L.

In addition, we also publicly recruited healthy participants over the age of 18, who are required to have no history of smoking or drinking, no history of chronic diseases, and a BMI within the normal range. And it was confirmed through physical examination that they were in a healthy state.

Seventy-six participants were recruited in this study, including 16 healthy controls (HC), 17 patients with viral pneumonia complicated with diabetes (VD), and 43 patients with viral pneumonia without diabetes (VP). All participants signed informed consent forms.

## Sample collection and processing

The sputum samples were collected from VP and VD patients under supervision. Following oral hygiene with sterile water, deep-cough specimens were expectorated into sterile containers. The sputum was kept in a sterile sampling cup and stored at −80°C. After thawing the sputum, liquefy it. Add 1.5 mL of 0.5% DTT, invert and mix about 10 times, and let it sit for 15–30 min. During the incubation process, invert and mix several times. Use a straw to divide liquefied sputum, approximately 1.6 mL–1.8 mL per cryovial. After labeling, store in a −80°C freezer. We collected 2.5 mL fasting peripheral venous blood samples from individuals in the VP group, VD group, and HC group using PAXgene tubes. When using a needle for blood collection, it is necessary to remove any air bubbles from the tail tube of the butterfly wing needle. After blood collection, immediately invert the blood collection tube 10–12 times, let it stand at room temperature for 2 h (not exceeding 8 h), and then transfer it to a −80°C refrigerator for storage.

## mRNA sequencing and transcriptome analysis

Sequencing was performed on 43 blood samples extracted from 43 participants. Total RNA was isolated using the standard procedures of the PAXgene Blood RNA Kit (Qiagen, Germany). RNA integrity was assessed using the RNA Nano 6000 Assay Kit of the Bioanalyzer 2100 system (Agilent Technologies, CA, USA). Qualified RNA was subjected to mRNA sequencing in Novogene Co., Ltd., and an average of 6 Gb of raw data was generated per sample. The index of the reference genome was built using Hisat2 v.2.0.5, and paired-end clean reads were aligned to the reference genome using Hisat2 v.2.0.5. The FeatureCounts v.1.5.0-p3 was used to count the read numbers mapped to each gene. And then, the FPKM of each gene was calculated based on the length of the gene and the read count mapped to it. Differential expression analysis between groups was performed using the DESeq2 R package (1.20.0). Genes with an adjusted *P*-value <0.05, as determined by DESeq2, were assigned as differentially expressed. Gene Ontology (GO) enrichment analysis of differentially expressed genes was implemented using the clusterProfiler R package, in which gene length bias was corrected. We used the clusterProfiler R package to test the statistical enrichment of differentially expressed genes in KEGG pathways (11).

## 16S rRNA gene sequencing

We performed 16S rRNA and metagenomic sequencing on all sputum samples separately. PCR amplification of the bacterial 16S rRNA genes V3–V4 region was performed using the forward primer 338F (5′-ACTCCTACGGGAGGCAGCA-3′) and the reverse primer 806R (5′-GGACTACHVGGGTWTCTAAT-3′). The PCR components contained 5 µL of buffer (5×), 0.25 µL of Fast pfu DNA Polymerase (5 U/µL), 2 µL (2.5 mM) of dNTPs, 1 µL (10 µM) of each forward and reverse primer, 1 µL of DNA template, and 14.75 µL of ddH$_2$O. Thermal cycling consisted of initial denaturation at 98°C for 5 min, followed by 25 cycles consisting of denaturation at 98°C for 30 s, annealing at 53°C for 30 s, and extension at 72°C for 45 s, with a final extension of 5 min at 72°C. PCR amplicons were purified with Vazyme VAHTSTM DNA Clean Beads (Vazyme, Nanjing, China) and

quantified using the Quant-iT PicoGreen dsDNA Assay Kit (Invitrogen, Carlsbad, CA, USA). After the individual quantification step, amplicons were pooled in equal amounts, and pair-end 2 × 250 bp sequencing was performed using the Illumina NovaSeq platform with the NovaSeq 6000 SP Reagent Kit (500 cycles) at Shanghai Personal Biotechnology Co., Ltd. (Shanghai, China), and an average of 60,000 raw reads was obtained per sample. Vsearch was used to perform quality filtering, merge paired-end reads, and cluster sequences into 97% operational taxonomic units (OTUs).

We used Vsearch and R (v.4.2.2) to analyze sequence data. We calculated the Chao1 richness estimator, observed species, Shannon diversity index, and Simpson index. We also plotted the richness curve of OTUs. Beta diversity analysis was performed by calculating the Jaccard index, the Bray-Curtis index, and the UniFrac distance index to compare the between-group changes in microbial communities.

## Metagenomic sequencing and analysis

Total microbial genomic DNA samples were extracted using the OMEGA Mag-Bind Soil DNA Kit (M5635-02) (Omega Bio-Tek, Norcross, GA, USA). Subsequently, the quantity and quality of DNA were measured. The extracted microbial DNA was processed to construct metagenome shotgun sequencing libraries with insert sizes of ~400 bp using the Illumina TrueSeq Nano DNA LT Library Preparation Kit (Illumina, USA). Libraries were sequenced on an Illumina platform, and for metagenomic sequencing, an average of 10 Gb of raw data was generated per sample. Raw sequencing reads were processed to obtain quality-filtered reads for further analysis. Once quality-filtered reads were obtained, taxonomical classifications of metagenomic sequencing reads from each sample were performed using Kraken2 (v.2.0.8-beta) against k2_pluspf_20240112 database. Reads assigned to metazoans or viridiplantae were removed fordownstream analysis. LEfSe and DESeq2 were used to identify differentially abundant taxa across groups. Filtered reads from each sample were assembled using MEGAHIT (v1.1.2) with the "--meta-large" preset. Generated contigs (>300 bp) were pooled and clustered using MMseqs2 (v13.45111) with the "easy-linclust" mode at 95% sequence identity and 90% coverage, retaining the longest representative sequences as a non-redundant contig set. Gene prediction on the non-redundant contigs was performed using Prodigal (v2.6.3). Predicted protein sequences were clustered using MMseqs2 "easy-cluster" at 95% identity and 90% coverage to generate a non-redundant gene catalog. To assess gene abundances, high-quality reads from each sample were mapped to the gene catalog using Minimap2 (with parameters "-ax sr --sam-hit-only") and quantified with featureCounts, yielding read counts per gene per sample. Functional annotation of the non-redundant genes was performed by aligning them against the KEGG database using MMseqs2 "search" mode.

## Constructing a correlation network between microbiota and genes

Based on the abundance of microorganisms at the genus and species levels, Spearman analysis with FDR correction was used to explore the correlation between microbial and host gene expression. The cutoff for demonstrating the correlation between microorganisms and genes was an absolute value of the correlation $r$-value greater than or equal to 0.5, and $P_{adj}$ is less than 0.05.

We have uploaded the sequencing data to the National Genomics Data Center database, from which access can be obtained upon request to the corresponding author (Bioproject: HRA013912).

## RESULT

### Baseline characteristic

There were 76 participants in all for this study. We included 16 patients in the HC group, 17 patients in the VD group, and 43 patients in the VP group. Among patients in HC, there were 11 males (68.75%) and 5 females (31.25%); the mean age was 33.625 years,

with a standard deviation of 2.819 years. The BMI of HC was 22.825, with a standard deviation of 0.593. For those in VD, there were 8 males (47.06%) and 9 females (52.94%), with a mean age of 63.765 years (± 4.415) and a BMI of 25.23 (± 1.019). In the VP group, there were 22 males (51.16%) and 21 females (48.84%), with an overall mean age of 57.61 years (± 3.041) and a mean BMI of 24.874 (± 0.648). There was no statistical difference in gender ($\chi^2$ test, $P = 0.442$) or BMI (one-way ANOVA, $P = 0.142$) between the patients in the three groups. The age of HC is younger than that of the VD and VP groups (Kruskal-Wallis $H$ test, $P < 0.001$). All VD and VP patients provided sputum specimens. There were no statistically significant differences in age, gender, or BMI between the VP group and the VD group, and there was no significant difference in the prevalence of comorbidities such as asthma, chronic obstructive pulmonary disease, hypertension, and hyperlipidemia between the two groups (VD and VP) (Fisher's exact test, $P > 0.05$). Additionally, no significant differences were observed in the types of viral infections present in either group (Fisher's exact test, $P > 0.05$). The risk of coronary heart disease in the VD group is greater than that in the VP group (Fisher's exact test, $P = 0.04$). Compared to VP, VD patients have longer hospitalizations (Mann-Whitney $U$-test, $P = 0.009$) (Table 1).

In addition, we collected blood samples from 43 participants, including 10 from VD, 17 from VP, and all members of the HC group. For those in VD, there were 5 males (50.00%) and 5 females (50.00%), with a mean age of 59.800 years (± 16.342) and a BMI of 25.693 (± 4.185). In the VP group, there were 5 males (29.41%) and 12 females (70.59%), with an overall mean age of 65.118 years (± 15.206) and a mean BMI of 24.446 (± 4.364).

**TABLE 1** Baseline characteristics of all participants[b,c]

| | HC (n = 16) | VD (n = 17) | VP (n = 43) | P-value[a] |
|---|---|---|---|---|
| Baselines | | | | |
| Male/Female | 11/5 | 8/9 | 22/21 | 0.442 |
| Age (mean ± SD) | 33.625 ± 2.819 | 63.765 ± 4.415 | 57.61 ± 3.041 | <0.001* |
| Height (mean ± SD) | 170 ± 2.006 | 164.647 ± 2.27 | 167.024 ± 1.168 | 0.168 |
| Weight (mean ± SD) | 66.388 ± 2.808 | 68.471 ± 3.21 | 69.722 ± 2.175 | 0.693 |
| BMI (mean ± SD) | 22.825 ± 0.593 | 25.23 ± 1.019 | 24.874 ± 0.648 | 0.142 |
| Comorbidities | | | | |
| Asthma, n (%) | /[d] | 2 (11.8%) | 0 (0%) | 0.136 |
| COPD, n (%) | / | 1 (5.8%) | 4 (9.3%) | 1 |
| Coronary heart disease, n (%) | / | 6 (35.3%) | 4 (9.3%) | 0.040* |
| Hypertension, n (%) | / | 5 (29.4%) | 7 (16.3%) | 0.431 |
| Hyperlipidemia, n (%) | / | 10 (58.8%) | 12 (27.92%) | 0.081 |
| Hospitalizations (mean ± SD) | / | 15.82 ± 9.26 | 9.83 ± 5.94 | 0.009* |
| Mortality rate, n (%) | / | 0 (0%) | 1 (5.8%) | 1 |
| Virus types | | | | |
| Novel coronaviruses, n (%) | / | 10 (23.3%) | 13 (76.5%) | 0.079 |
| Influenza A, n (%) | / | 5 (11.6%) | 10 (58.8%) | 0.869 |
| Adenovirus, n (%) | / | 1 (2.3%) | 10 (58.8%) | 0.231 |
| Respiratory syncytial virus, n (%) | / | 1 (2.3%) | 7 (41.2%) | 0.518 |
| Parapneumovirus, n (%) | / | 0 (0%) | 5 (29.4%) | 0.342 |
| Coronavirus, n (%) | / | 0 (0%) | 4 (23.5%) | 0.467 |
| Parainfluenza virus, n (%) | / | 1 (2.3%) | 2 (11.8%) | 1 |
| Human rhinovirus, n (%) | / | 1 (2.3%) | 2 (11.8%) | 1 |

[a]Statistical tests: the statistical analysis of categorical variables between three groups uses the $\chi^2$ test, and the statistical analysis of continuous variables between three groups uses one-way analysis of variance. If the data do not meet normality or homogeneity of variance, the Kruskal-Wallis $H$ test is used. Comparing categorical variables between two groups, Fisher's exact test was used due to the small sample size in this study. The independent sample $t$-test is used to compare continuous variables between two groups. If the variance is uneven or non-normal, the Mann-Whitney $U$-test is used.

[b]HC, healthy control group; VD, viral pneumonia with diabetes group; VP, viral pneumonia without diabetes group; COPD, chronic obstructive pulmonary disease.

[c]"*" indicates statistically significant difference ($P < 0.05$).

[d]/, healthy individuals without comorbidities or viral infections.

**TABLE 2** Basic information of participants providing blood samples[b]

| | HC (*n* = 16) | VD (*n* = 10) | VP (*n* = 17) | *P*-value[a] |
|---|---|---|---|---|
| Baselines | | | | |
| Male/Female | 11/5 | 5/5 | 5/12 | 0.276 |
| Age (mean ± SD) | 33.625 ± 2.819 | 59.800 ± 16.342 | 65.118 ± 15.206 | 0.283 |
| Height (mean ± SD) | 170 ± 2.006 | 167.600 ± 12.195 | 165.400 ± 5.962 | 0.339 |
| Weight (mean ± SD) | 66.388 ± 2.808 | 72.200 ± 14.920 | 67.067 ± 13.023 | 0.504 |
| BMI (mean ± SD) | 22.825 ± 0.593 | 25.693 ± 4.185 | 24.446 ± 4.364 | 0.264 |
| Comorbidities | | | | |
| Asthma, *n (%)* | /[c] | 0 (0%) | 0 (0%) | 1 |
| COPD, *n (%)* | / | 0 (0%) | 1 (5.8%) | 0.824 |
| Coronary heart disease, *n (%)* | / | 0 (0%) | 1 (5.8%) | 0.749 |
| Hypertension, *n (%)* | / | 6 (60.0%) | 6 (35.3%) | 0.386 |
| Hyperlipidemia, *n (%)* | / | 5 (50.0%) | 5 (29.4%) | 0.309 |
| Hospitalizations (mean ± SD) | / | 12.00 ± 6.18 | 14.60 ± 10.99 | 0.711 |
| Mortality rate, *n (%)* | / | 0 (0%) | 0 (0%) | 1 |
| Virus types | | | | |
| Novel coronaviruses, *n (%)* | / | 8 (80.0%) | 9 (52.94%) | 0.264 |
| Influenza A, *n (%)* | / | 0 (0%) | 2 (11.76%) | 0.639 |
| Adenovirus, *n (%)* | / | 1 (10.0%) | 3 (17.65%) | 0.749 |
| Respiratory syncytial virus, *n (%)* | / | 1 (10.0%) | 3 (17.65%) | 0.749 |
| Parapneumovirus, *n (%)* | / | 0 (0%) | 0 (0%) | 1 |
| Coronavirus, *n (%)* | / | 1 (10.0%) | 2 (11.76%) | 0.824 |
| Parainfluenza virus, *n (%)* | / | 0 (0%) | 1 (5.88%) | 0.941 |
| Human rhinovirus, *n (%)* | / | 0 (0%) | 2 (11.76%) | 0.639 |

[a]Statistical tests: the statistical analysis of categorical variables between three groups uses the $\chi^2$ test, and the statistical analysis of continuous variables between three groups uses one-way analysis of variance. If the data do not meet normality or homogeneity of variance, the Kruskal-Wallis *H* test is used. Comparing categorical variables between two groups, Fisher's exact test was used due to the small sample size in this study. The independent sample *t*-test is used to compare continuous variables between two groups. If the variance is uneven or non-normal, the Mann-Whitney *U*-test is used.
[b]HC, healthy control group; VD, viral pneumonia with diabetes group; VP, viral pneumonia without diabetes group; COPD, chronic obstructive pulmonary disease.
[c]/, healthy individuals without comorbidities or viral infections.

The basic information of the HC group is the same as above. There were no significant differences in age, gender, height, BMI, other comorbidities, mortality rates, or types of viral infections (Table 2).

## Transcriptome profiling

Principal component analysis was performed on the plasma transcriptome data of the HC, VD, and VP groups; the contribution of PC1 and PC2 was 22 and 14.6%. It can be seen that VD and VP are separated from HC to a certain extent. There is also a certain separation between VD and VP (Fig. 1a). For further analysis, we identified differentially expressed genes between the VD group and VP group, HC group, and all patients with viral pneumonia (including VD and VP groups). When compared to healthy controls, 1,224 genes were upregulated and 705 genes were downregulated in patients with viral pneumonia. Compared with the VP group, the VD group had 248 genes upregulated and 290 genes downregulated. The Venn diagram shows the simultaneous upregulation or downregulation of gene expression in pairwise comparisons (Fig. 1b). The volcano map also shows that there are differentially expressed genes in the VD group compared to the VP group (Fig. 1c). Then, we paid special attention to genes related to diabetes and cell growth or development: compared with the VP group, the VD group downregulated the expression of PTCH1 and upregulated the expression of ANK1, RBM38, BPGM, CRYM, TAL1, and HBD (Fig. 1d). Functionally, PTCH1 is a receptor for the Hedgehog signaling pathway, which is involved in regulating cell proliferation, differentiation, and tissue formation (12). ANK1, TAL1, and HBD are involved in the maintenance of red blood cells

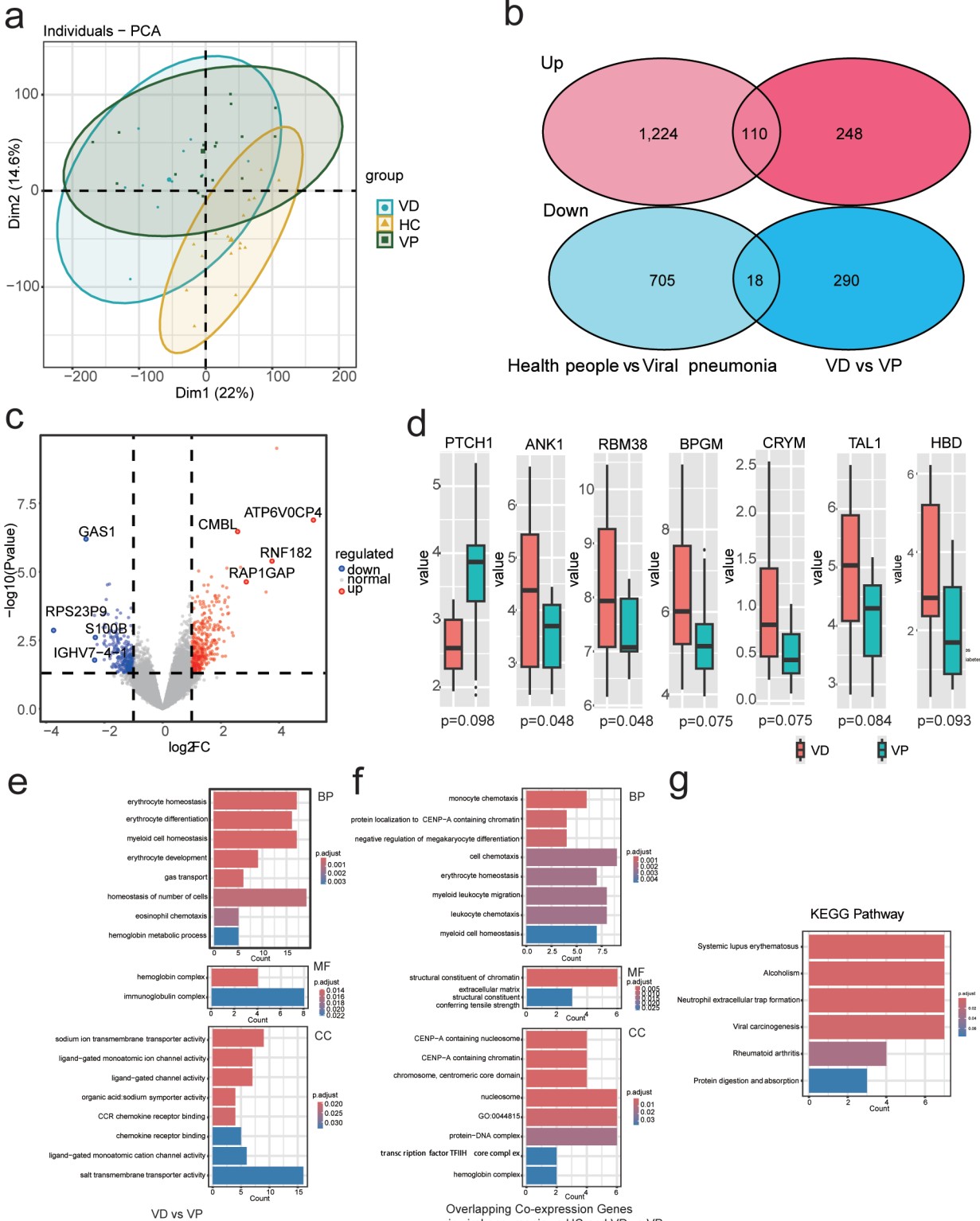

FIG 1  Differentially expressed genes analysis between VD, VP, and HC groups. (a) Principal component analysis shows dissimilarity between the VD, VP, and HC groups. (b) Venn diagram of differentially expressed genes in HC compared with all patients with viral pneumonia, and VD compared to VP. (c) Volcano plot of differentially expressed genes for VD vs VP. The genes with significant differences in expression ($P < 0.05$) are highlighted in red (upregulated) and blue (downregulated). (d) Box plot of differentially expressed genes between VD and VP. (e) GO pathways for differentially expressed genes for VD vs VP. (f) GO pathways of co-expressed genes in the two comparison groups. (g) KEGG pathways of co-expressed genes in two comparison groups.

and hemoglobin, as well as the development of hematopoietic stem cells (13, 14). BPGM and CRYM are of great significance for energy metabolism in the body, and RBM38 can affect cell survival by regulating the expression of target genes (15).

## Diabetes may be related to the suppression of immune and metabolic pathways in VD

For the differential expression between the VD group and the VP group, GO enrichment analysis found that the difference was mainly reflected in the formation, development, and maintenance of red blood cells, the activity of immunoglobulin, and the transmembrane transport of substances (Fig. 1e). In the previous step, we analyzed the differential genes of viral pneumonia patients and healthy people and the differential genes of the VD group compared with the VP group. We carried out GO enrichment analysis on the genes expressed in the same direction in the comparison of the two groups. Genes related to the movement and migration of immune cells, the stability of myeloid cells, and DNA damage repair pathways were simultaneously downregulated in the two comparisons (Fig. 1f). Co-expressed genes are primarily linked to immunological and metabolic pathways, including systemic lupus erythematosus, alcoholism, neutrophil extracellular trap formation, viral carcinogenesis, rheumatoid arthritis, and protein digestion and absorption pathways, according to KEGG enrichment analysis (Fig. 1g).

## Diabetes may be related to the decline of respiratory microbial diversity and dysfunction in VD patients

We conducted both 16S rRNA and metagenome analysis on 60 sputum samples separately, which suggested that the results of alpha diversity showed a significant increase in the VP group compared to those in the VD group (Wilcoxon test, $P = 0.04$) (Fig. 2a). This suggests that the respiratory microbiota diversity is higher in the VP group than in the VD group. We used PCoA for visualization, and the PerMANOVA test based on Bray-Curtis showed that there were significant differences in the microbial community structure between the two groups (PerMANOVA test, $P = 0.04$) (Fig. 2b). At the family classification level, the VD group did not differ significantly from the VP group and was mainly composed of the family *Streptococcaceae*, the family *Veillonellaceae*, the family *Prevotellaceae*, the family *Micrococcaceae*, and the family *Actinomycetaceae* (Fig. 2c). At the genus level, the top 5 bacterial genera in both groups were *Streptococcaceae*, *Rothia*, *Actinomyces*, *Veillonella*, and *Prevotella* (Fig. 2d).

The differences in different levels of bacterial flora between the VD and VP groups were analyzed by DESeq2 based on a negative binomial distribution model, which showed that *g__Mastadenovirus*, *g__Treponema*, *g__Selenomonas*, *g__Haemophilus*, *g__Lautropia*, *g__Mogibacterium,* and *g__Gemella* increased, and *s__Treponema_denticola* and *s__Campylobacter_rectus* increased in the VP group, while *g__Tannerella*, *g__Bifidobacterium*, *g__Lacticaseibacillus*, *g__Enterococcus*, and *g__Staphylococcus* increased in the VD group (DESeq2 Wald test, $P < 0.05$). LEfSe was used to further analyze the bacterial groups that were differentially enriched between the VD and VP groups and screened to obtain the groups with LDA scores > 2. *O__Spirochaetales*, *p__Spirochaetota*, *c__Spirochaetia*, *f__Treponemataceae*, *g__Treponema*, *s__Campylobacter_rectus*, *o__Selenomonadales*, *f__Selenomonadaceae,* and *s__Treponema_denticola* were significantly enriched in the VP group (Kruskal-Wallis *H* test, $P < 0.05$). Both analysis methods demonstrate a significant increase in the abundance of *g__Treponema*, *s__Treponema_denticola,* and *s__Campylobacter_rectus* in the VP group (Fig. S1).

Further analysis of bacterial KEGG differential expression revealed significant differences in microbial functional expression between the two groups in cellular processes, environmental information processing, genetic information processing, disease-related processes, and metabolic processes (Fig. 2e). The VD group showed enrichment in the p53 signaling pathway, biofilm formation in Vibrio cholerae, and flagellar assembly expression. The p53 signaling pathway is associated with important processes such as cell cycle regulation and apoptosis, and its abnormalities may affect

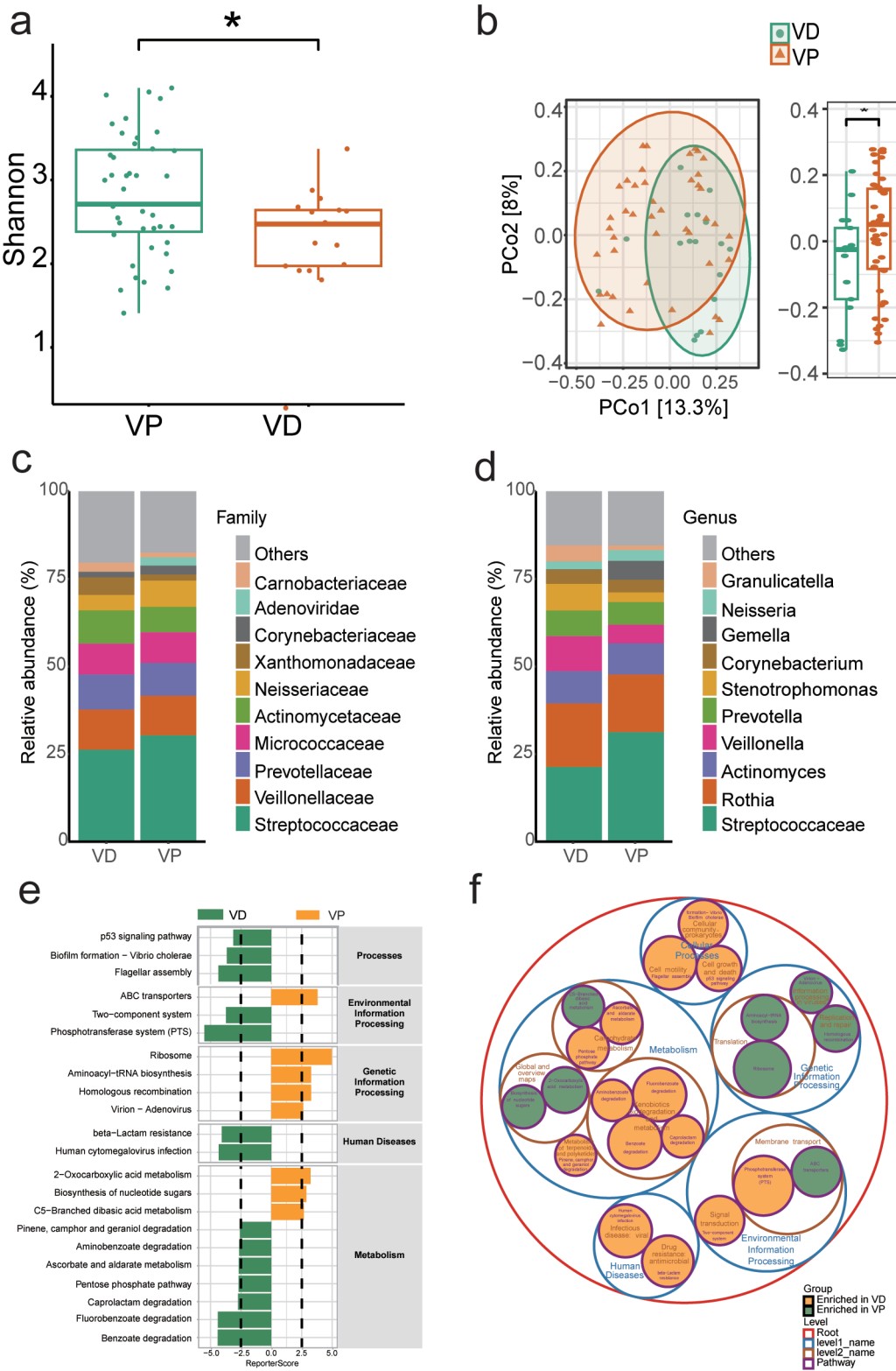

**FIG 2** (a) Shannon index of the VD and VP groups. (b) Principal coordinate analysis plot comparing the composition of microbiota in the VD and VP groups. (c) Analysis of differences in microbial communities between the two groups at the family level. (d) Analysis of differences in microbial communities between the two groups at the genus level. (e, f) KEGG pathways analyze microbial functional expression between the two groups. * Statistically significant difference ($P < 0.05$).

the normal function and metabolism of cells (16). We inferred from the inhibition of the expression of the latter two pathways that the biofilm formation and motility of microorganisms may be negatively affected in the state of diabetes. During environmental information processing, the ABC transporters in the VP group were more active, while the two-component system and phosphotransferase system in the VD group were inhibited. In terms of genetic information processing, the VP group exhibited more active ribosome processes, aminoacyl tRNA biosynthesis, homologous recombination, and virion adenovirus. The beta-lactam resistance and human cytomegalovirus infection pathways were inhibited in the VD group. In terms of material metabolism, the degradation of pinene, camphor, aminobenzoate, caprolactam, fluorobenzoate, and geraniol in the VD group was inhibited. The circular chart visually displays the differences in microbial pathway enrichment between VD and VP groups at different levels (Fig. 2f).

## Interaction between transcriptome and microbiome

In order to further investigate the host-microbe relationship, we conducted co-occurrence analysis on microbes and transcriptome genes and obtained an association diagram as shown in the figure. The diagram consists of two main parts, reflecting the correlations between microbes and their potential associations with specific genes at the genus (Fig. 3a) and species (Fig. 3b) levels. The figure illustrates the complex interrelationships among different microbial species, where the blue or red blocks represent the synergistic or antagonistic effects exhibited by microorganisms. By associating the genes with the colonies, we found that RBM38, RAP1GAP, TMCC2, KANK2, CMBL, KDM7A–DT, ANK1, AGAP1, and RNF182 had more significant associations at the genus level. RBM38, ANKRD9, RAP1GAP, KANK2, TMCC2, CMBL, ANK1, RNF182, KDM7A–DT, and AGAP1 had more significant associations at the species level. AGAP1, RNF182, and ANKRD9 were particularly well represented. The host transcriptome gene AGAP1 is positively correlated with *g_Treponema*, *g_Gemella*, *s_Veillonella_rogosae*, and *s_Bifidobacteria*. The host transcriptome gene RNF182 is negatively correlated with *g_Treponema*, *g_Haemophilus*, *g_Mastadenovirus*, *s_Neisseria_subflava*, *s_Neisseria_sp._oral_taxon_014*, and *s_Neisseria_elongata*. The host transcriptome gene ANKRD9 is negatively correlated with *g_Gemella*, *g_Treponema*, *s_Bifidobacterium_longum*, *s_Campylobacter_rectus*, *s_Treponema_denticola*, and *s_Tannerella_forsythia*.

## DISCUSSION

By comparing the baseline conditions of the VP group and VD group, we found that the frequency of VD combined with coronary heart disease (CHD) was higher than that of the VP group. This is consistent with previous research results (17). As an independent risk factor for coronary heart disease, diabetes patients have a higher prevalence of CHD, a larger range of coronary artery ischemia, and are more likely to have myocardial infarction and asymptomatic myocardial ischemia than non-diabetes patients (18). The VD group had a longer hospital stay compared to the VP group, which further confirms the longer course and poorer prognosis of VD.

PTCH1 is inhibited in VD. PTCH1 is an important switch necessary for T and B lymphatic development (19, 20), and is correlated with lung development, airway regeneration, and lung function (21). Inhibition of PTCH1 will affect the process of the host immune response and affect the repair of lung tissue and lung function. VD significantly increased the expression of diabetes susceptibility genes, including ANK1 and RBM38, two genes that have an impact on the host's energy metabolism processes (22–24). In addition, VD increased the expression of BPGM, HBD, and TAL1. BPGM can reduce the affinity of cells for oxygen (15). HBD and TAL1 are associated with red blood cells (25). These three genes are related to the transport and utilization of oxygen, and the increase in expression may affect the process of oxygen uptake by tissue cells. Thus, we inferred that VD may inhibit the immune regulatory response, affect cell energy metabolism and oxygen transport and utilization by regulating related gene pathways, thus affecting the prognosis of patients with viral pneumonia.

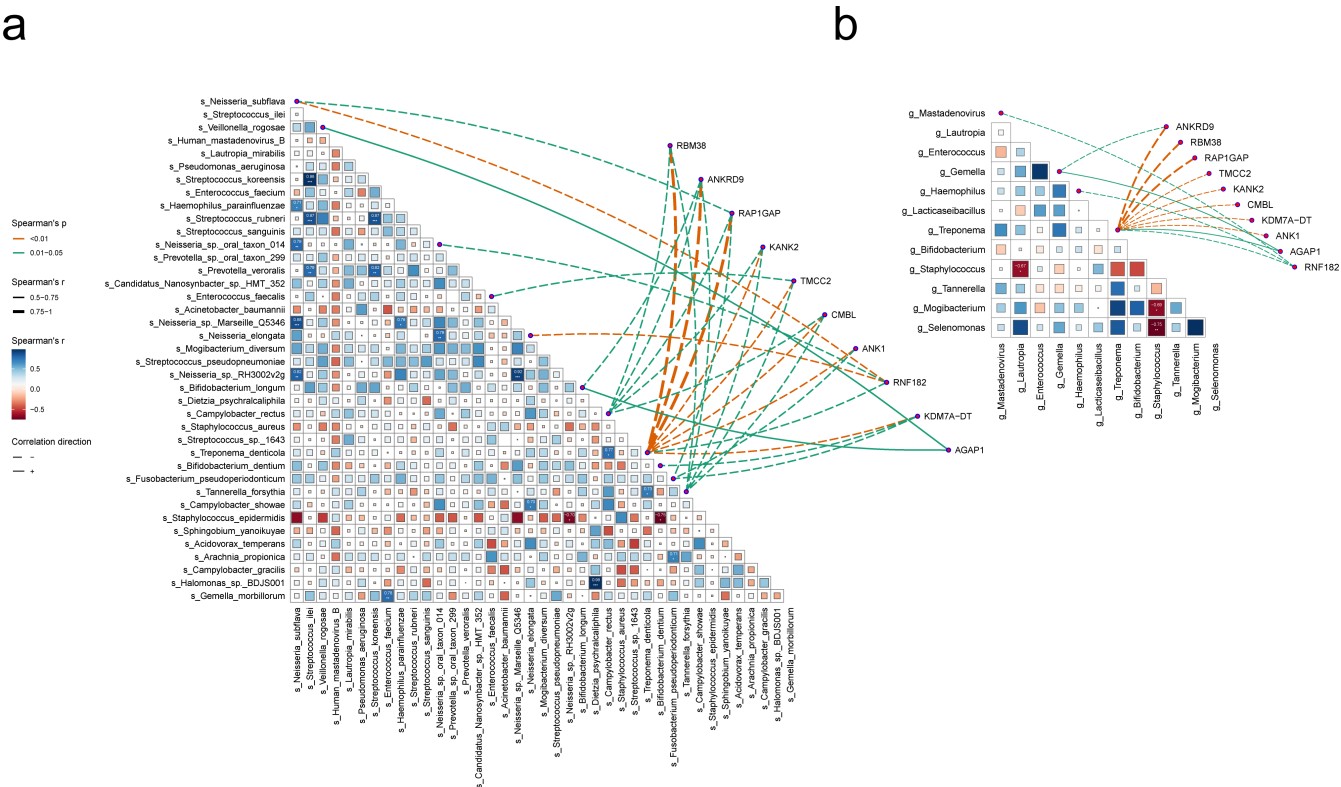

**FIG 3** Correlation analysis between microorganisms at the species (a) or genus (b) level and transcriptome genes. The blocks represent the correlation between microorganisms, with the color tending toward blue indicating a stronger positive correlation and the color tending toward red indicating a stronger negative correlation. The line connecting microorganisms and genes represents their relationship. If the line is a solid line, then the two are positively correlated, and if it is a dashed line, then they are negatively correlated.

Although there was no significant difference in family classification and genus level between VD and VP respiratory microorganisms, the diversity of the VD group was significantly reduced compared with the VP group. It is well known that the decrease in respiratory microbiota diversity will increase the risk of complications such as bacterial infection and fungal infection and hinder the regulation and repair of the respiratory tract. The abundance of *Treponema denticola* in the VP group was significantly higher than that in the VD group. *Treponema denticola* is an important oral bacterium involved in the occurrence of periodontal disease, especially in chronic periodontitis (26). *Treponema denticola* is associated with neurodegenerative diseases such as Alzheimer's disease by causing damage to host cells through its various virulence factors (27). A study shows that the abundance of oral microbiota *Treponema denticola* in type 2 diabetes may change (28). As a periodontal pathogen, *Treponema denticola* aggravates systemic inflammatory response by triggering periodontal inflammation and releasing inflammatory mediators, affecting the control of blood sugar in diabetes patients (29). The samples collected in this study are from the lower respiratory tract, so there may be differences in the trend of changes.

Further analysis of KEGG indicated that there were significant differences in microbial functional expression between the two groups. The p53 signaling pathway is a relevant pathway for host regulation of the cell cycle and maintenance of cell stability. Biofilm and flagella are influencing factors for microbial colonization and distribution in the host (30, 31). The above process pathways are all inhibited in the VD group. ABC transporters (32, 33), two-component systems (34, 35), and PTS (36) are all systems that process environmental information in microorganisms. The ABC transporters in the VP group are more active. The two-component system and phosphotransferase system of the VD group are inhibited, which may reduce the adaptability of VD group microorganisms to

environmental changes. The VP group has more active functions in ribosome, amino-acyl tRNA biosynthesis, homologous recombination, and virion adenovirus, reflecting a more vigorous microbial metabolism and genetic activity in the VP group. Significant metabolic disorders were observed in the VD group. Related substances are involved in host immune regulation, energy metabolism, and substance synthesis and can provide nutrients for specific microorganisms (37–39). We predicted that the microbial function of VD is inhibited in processes, environmental information processing, human diseases, and metabolism, which may further affect the host's immune regulation and substance transport function.

In traditional Chinese medicine, the deficiency of yin in diabetes patients cannot restrict yang, and the deficiency fire is hyperactive (40). Diabetes patients with yin deficiency with dry heat are often accompanied by poor blood sugar control, increased oxidative stress, and enhanced inflammatory response (41). The internal disturbance of internal organs by dry heat can easily lead to inflammation and immune and metabolic disorders. Dry heat burns the body's energy, causing weakness and affecting functions such as metabolism and oxygen utilization. Yin deficiency, internal heat generation, may have an impact on the living environment of microbiota (42). Our research found that gene expression related to immune regulation, cellular energy metabolism, and oxygen transport is inhibited in VD patients, while microbial composition and function undergo changes. This is consistent with traditional Chinese medicine's understanding of "yin deficiency and fire excess" and also interprets the pathological characteristics of "yin deficiency and fire excess" at the micro level.

To better understand the relationship between the microbiome and the host transcriptome, we performed a correlation analysis. AGAP1 is closely related to glucose uptake in diabetic patients (43, 44). Research has found that RNF182 exerts tumor suppressive effects in non-small cell lung cancer by inhibiting cell growth and promoting cell cycle arrest (45), and may be associated with idiopathic pulmonary fibrosis (46). ANKRD9 is involved in various biological processes such as protein assembly, cytoskeletal interactions, post-translational modifications, and protein degradation (47). It plays an important role in lipid metabolism, copper homeostasis, and cell proliferation (48). ANKRD9 may be involved in building a bridge between host gene expression and microbial metabolic dysfunction in VD patients. AGAP1, RNF182, and ANKRD9 may be the pathway of host-respiratory microbial interaction in the VD group, and we can continue to carry out relevant studies in the future.

Noteworthy, due to social factors such as an aging population, the population of hospitalized patients is generally older, and the age of healthy individuals is younger than that of VD and VP patients. Although there was no significant difference in age between VD and VP groups, this may affect the analysis of transcriptome differentially expressed genes, which is also one of the limitations of this study. In addition, although the patient does not have diabetes, he may have other systemic diseases, which may affect our transcriptome analysis results. We used a combination of multiple omics to discover the relationship between the host transcriptome and microbiome, which still requires further research. Other researchers can further explore and develop corresponding microbial regulators in clinical practice to improve the condition of viral pneumonia patients with diabetes and shorten the disease process by targeting microorganisms and pathways.

## Conclusion

Our microbiome and transcriptome results indicate that the long course and poor prognosis of patients with viral pneumonia and diabetes may be related to two aspects. The expression of genes related to immune regulation, energy metabolism, and oxygen utilization in patients with viral pneumonia and diabetes is abnormal. At the same time, we predicted that the respiratory tract microbiota shows abnormal metabolic activity, diversity, and adaptability to the environment. Further investigation of the mechanism suggests that AGAP1, RNF182, and ANKRD9 may be the pathways between host and

respiratory microbial interaction. We also provide a certain mechanism interpretation for yin deficiency with dry heat in "Xiao ke."

## ACKNOWLEDGMENTS

Gratitude to the teachers of the sample library and the patients who participated in the study.

This study was supported by the National Key Research and Development Program of China (nos. 2022YFC3500802 and 2022YFA1304303), the National High Level Hospital Clinical Research Funding (2025-NHLHCRF-JBGS-B-WZ-06), Beijing Research Ward Excellence Program (BRWEP2024W114060104), the Shenzhen Medical Research Fund (C2401006), and the China-Japan Friendship Hospital Elite Program (ZRJY2023-GG24).

## AUTHOR AFFILIATIONS

[1]Beijing University of Chinese Medicine, Beijing, China

[2]National Center for Respiratory Medicine; National Centre for Integrative Chinese and Western Medicine; State Key Laboratory of Respiratory Health and Multimorbidity, National Clinical Research Center for Respiratory Diseases; Institute of Respiratory Medicine, Chinese Academy of Medical Sciences; Department of Traditional Chinese Medicine for Pulmonary Diseases, China-Japan Friendship Hospital, Beijing, China

[3]Tsinghua University Yuquan Hospital (Tsinghua University Hospital of Integrated Traditional Chinese and Western Medicine), Beijing, China

[4]National Center for Respiratory Medicine; State Key Laboratory of Respiratory Health and Multimorbidity, National Clinical Research Center for Respiratory Diseases; Institute of Respiratory Medicine, Laboratory of Clinical Microbiology and Infectious Diseases, Department of Pulmonary and Critical Care Medicine, Center of Respiratory Medicine, Chinese Academy of Medical Sciences, China-Japan Friendship Hospital, Beijing, China

## AUTHOR ORCIDs

Hongchun Zhang http://orcid.org/0009-0002-8523-3139
Xiaohui Zou http://orcid.org/0000-0003-0607-4759

## FUNDING

| Funder | Grant(s) | Author(s) |
| --- | --- | --- |
| National Key Research and Development Program of China | No2022YFC3500802, 2022YFA1304303 | Xiaohui Zou |
| National High Level Hospital Clinical Research Funding | 2025-NHLHCRF-JBGS-B-WZ-06 | Xiaohui Zou |
| China-Japan Friendship Hospital Elite Program | ZRJY2023-GG24 | Xiaohui Zou |
| Beijing Research Ward Excellence Program | BRWEP2024W114060104 | Xiaohui Zou |
| Shenzhen Medical Research Fund | C2401006 | Xiaohui Zou |

## AUTHOR CONTRIBUTIONS

Changrui Huang, Data curation, Formal analysis, Writing – original draft, Writing – review and editing | Qinqi Feng, Data curation, Formal analysis | Bang Yu, Methodology | Hao Zou, Supervision | Yashi Cai, Validation, Visualization | Jian Liu, Project administration | Demin Li, Project administration | Hongchun Zhang, Project administration, Supervision, Writing – review and editing | Xiaohui Zou, Project administration

## DATA AVAILABILITY

The raw sequence data reported in this paper have been deposited in the Genome Sequence Archive (Genomics, Proteomics & Bioinformatics 2025) in the National Genomics Data Center (Nucleic Acids Res 2025), China National Center for Bioinformation/Beijing Institute of Genomics, Chinese Academy of Sciences (GSA-Human: HRA013912), which can be applied for access from the corresponding author.

## ETHICS APPROVAL

This study was derived from a prospective cohort study of patients with viral pneumonia, and the clinical trial was approved by the Ethics Committee of the China-Japan Friendship Hospital (approval number: 2023-KY-154).

## ADDITIONAL FILES

The following material is available online.

### Supplemental Material

**Figure S1 (Spectrum01911-25-s0001.tif).** Differences in microorganisms between VD and VP Groups.
**Supplemental legends (Spectrum01911-25-s0002.docx).** Legends for Tables S1 to S7 and Figure S1.
**Table S1 (Spectrum01911-25-s0003.xlsx).** Kingdom-level species abundance table.
**Table S2 (Spectrum01911-25-s0004.xlsx).** Phylum-level species abundance table.
**Table S3 (Spectrum01911-25-s0005.xlsx).** Class-level species abundance table.
**Table S4 (Spectrum01911-25-s0006.xlsx).** Order-level species abundance table.
**Table S5 (Spectrum01911-25-s0007.xlsx).** Family-level species abundance table.
**Table S6 (Spectrum01911-25-s0008.xlsx).** Genus-level species abundance table.
**Table S7 (Spectrum01911-25-s0009.xlsx).** Species-level species abundance table.

### Open Peer Review

**PEER REVIEW HISTORY (review-history.pdf).** An accounting of the reviewer comments and feedback.

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
