## [Reviewer comments · Microbiology Spectrum]

Microbiology Spectrum

Diabetes Affects the Composition of the Respiratory Tract Microbiome and Transcriptome in Patients with Viral Pneumonia

Changrui Huang, Qinqi Feng, Yashi Cai, Hao Zou, Bang Yu, Jian Liu, Demin LI, Xiaohui Zou, and Hongchun Zhang

Corresponding Author(s): Xiaohui Zou, China-Japan Friendship Hospital

Review Timeline:

Submission Date:	June 29, 2025
Editorial Decision:	September 27, 2025
Revision Received:	October 23, 2025
Editorial Decision:	November 20, 2025
Revision Received:	November 21, 2025
Accepted:	January 28, 2026

Editor: Rup Lal

Reviewer(s): Disclosure of reviewer identity is with reference to reviewer comments included in decision letter(s). The following individuals involved in review of your submission have agreed to reveal their identity: Mingkun Li (Reviewer #1); Hui Wen (Reviewer #2)

Transaction Report:

DOI: <https://doi.org/10.1128/spectrum.01911-25>

Re: Spectrum01911-25 (**Diabetes Affects the Composition of the Respiratory Tract Microbiome and Transcriptome in Patients with Viral Pneumonia**)

Dear Dr. Xiaohui Zou:

Thank you for the privilege of reviewing your work. Below you will find my comments, instructions from the Spectrum editorial office, and the reviewer comments.

Revision Guidelines

Sincerely,
Rup Lal
Editor
Microbiology Spectrum

Reviewer #1 (Comments for the Author):

Comments to the Author

The authors enrolled 76 participants, divided them into three group, healthy control, VD, and VP. Sputum and blood samples were collected for metagenomic and RNA sequencing, revealing diabetes influences both the microbial composition and host gene expression in patients with viral pneumonia. The conclusion of this article has certain value for the treatment of patients

with diabetic pneumonia. The research method and conclusion are generally reliable.

Comments

1. The Methods section lacks sufficient detail and should be expanded. For example, in "Study Population", please clearly specify the total number of participants, the number in each group, and the inclusion criteria for each group. And in Line 149, details regarding blood and sputum sample collection - including the collection methods, materials used, and sample volumes - are missing and should be clearly described.
2. Figures corresponding to the results should be clearly referenced in the main text, indicating which figure corresponds to which result. And some results lack proper statistical testing. For instance, the statement on Lines 240-245, "It was found that there were significant differences in gene expression among the three groups," needs supporting statistical analysis. The authors should report the specific statistical tests used and their corresponding p-values to validate this claim.
3. The study reports collecting 76 sputum samples, but only 60 were sequenced. Please clarify how these 60 samples were selected. In addition, it is unclear whether the microbiome analysis was performed using 16S rRNA sequencing or metagenomic sequencing.
4. In Fig. 3, it is not clear whether the reported correlations between microbes and genes are positive or negative. Do the microbes promote or suppress the expression of these genes? What are the potential biological mechanisms underlying these associations? At present, the results appear to report raw correlation p-values without correction for multiple testing (q-values), making it likely that many of these correlations arise by chance. In its current form, the findings offer limited insight into the true relationship between the microbiome and the transcriptome.
5. Line 66: The manuscript mentions that "survival rates are reduced", but no clinical outcome data are provided in the results to support this statement.
6. The Introduction section briefly mentions "traditional Chinese medicine", but the manuscript as a whole does not appear to include any analyses or discussion related to Traditional Chinese Medicine. This reference should either be expanded upon or removed for clarity.
7. Sequencing data should be publicly available.
8. Line 282: a typo in "16S RNA and birdshot macrogenomic analyses".

Reviewer #2 (Comments for the Author):

This manuscript examines how diabetes influences host transcriptomic responses and respiratory microbiome composition in patients with viral pneumonia, based on integrated analyses of blood and sputum samples from 76 participants. The authors report that diabetes is associated with significant alterations in both host transcriptomic profiles and respiratory microbiome characteristics, including reduced microbial diversity, distinct community composition, and changes in metabolic pathways. Overall, the manuscript is clearly structured, with well-described methods and logically presented results that are clinically relevant. In parts, interpretation could be improved (see comments below).

1. The sample size is modest, particularly for the transcriptomic analyses (n=27 patients), which limits power to detect subtle differences. Also, the age distribution differs substantially between healthy controls and patient groups, which may confound results. While some comparisons are made between VP and VD, the controls are still considerably younger, complicating the interpretation of "healthy vs. diseased" contrasts. These limitations should be acknowledged more explicitly.
2. The reliance on LEfSe with LDA thresholds can sometimes overstate significance when analyzing microbial differences. Have the authors considered using alternative methods (e.g., PERMANOVA) to strengthen the robustness of the α - and β -diversity comparisons?
3. The correlation network between transcriptomic and microbial features is interesting, but the statistical thresholds for inclusion (e.g., correlation cutoffs, multiple testing control) are not described in detail. Could the authors provide more information about this?
4. It is not quite clear if the comorbidities were adjusted for in transcriptome or microbiome analyses, while this is reported in Table 1 (e.g., coronary heart disease was more common in the VD group).
5. Some conclusions are stated too strongly given the cross-sectional design, which limits causal inference (e.g., statements that diabetes "inhibits" immune regulation or microbial adaptability). Because the analyses rely on inferred functional enrichment from metagenomic data, it would be more accurate to conclude these findings as associations or predicted functional differences rather than definitive mechanistic effects.
6. The integration of transcriptomic and microbiome findings with traditional Chinese medicine concepts is interesting. However, the link between molecular results and TCM interpretation is sometimes asserted rather than explained. The authors may wish to provide a clearer rationale for how specific gene or microbial changes map onto TCM constructs.
7. Given the large number of differentially expressed genes and microbial taxa, consider providing full results (e.g., DESeq2

output, taxa abundance tables) as supplementary material for transparency and reproducibility.

8. Please provide a detailed description of the three figures (and titles), and make sure that all panels are explicitly referenced in the main text (e.g., Figure 1a). It's difficult for readers to follow the connection between results and visuals.

9. In Table 1, the entry for "Hospitalizations (Mean {plus minus} SD)" shows a p-value of 9.547. Please verify the calculation and correct the value (or, if 9.547 is a test statistic, report it in footnote separately and provide the corresponding p-value). Also, please specify in the table footnotes the statistical tests used for each variable in Tables 1 and 2.

10. The formatting of Tables 1 and 2 is not fully consistent (e.g., some variables are reported with two decimals while others are not; some values are center-aligned while others are left-aligned; one table bolds some rows and the other does not; and text fonts appear mismatched). p-value for Height in Table 2 appears to have a typo (double leading 0).

Dear Editor,

On behalf of all the authors, I would like to thank you and all the reviewers for your constructive comments and suggestions on our manuscript. The manuscript has been revised comprehensively according to the editor's and reviewers' suggestions/comments. We also have the manuscript reviewed. Below are the details of how we have addressed the critiques of all the reviewers/editors.

Reviewer #1:

The authors enrolled 76 participants, divided them into three group, healthy control, VD, and VP. Sputum and blood samples were collected for metagenomic and RNA sequencing, revealing diabetes influences both the microbial composition and host gene expression in patients with viral pneumonia. The conclusion of this article has certain value for the treatment of patients with diabetic pneumonia. The research method and conclusion are generally reliable.

Answer: Thanks for your kindly suggestion. We have carefully revised and streamlined the content of the manuscript, and the chart content is also more intuitive.

Some specific comments follow:

1) The Methods section lacks sufficient detail and should be expanded. For example, in "Study Population", please clearly specify the total number of participants, the number in each group, and the inclusion criteria for each group. And in Line 149, details regarding blood and sputum sample collection - including the collection methods, materials used, and sample volumes - are missing and should be clearly described.

Answer: Thanks. We have clearly stated the total number of participants, the number of participants in each group, and the inclusion criteria for each group in the Methods and Results section. Please see it in our revised manuscript Lines 143-158. And we also added details on the collection of blood and sputum samples in manuscript Lines 160-173.

2) Figures corresponding to the results should be clearly referenced in the main text, indicating which figure corresponds to which result. And some results lack proper statistical testing. For instance, the statement on Lines 240-245, "It was found that there were significant differences in gene expression among the three groups," needs supporting statistical analysis. The authors should report the specific statistical tests used and their corresponding p-values to validate this claim.

Answer: Thanks for suggestion. We have added the cited images at the corresponding positions in the manuscript. We reported the P-value of the statistical analysis and corrected some inappropriate expressions (Lines 274-276).

3) The study reports collecting 76 sputum samples, but only 60 were sequenced. Please clarify how these 60 samples were selected. In addition, it is unclear whether the microbiome analysis was performed using 16S rRNA sequencing or metagenomic sequencing.

Answer: Thanks for your questions. Sorry, due to a typo, the description was incorrect earlier. A total of 60 sputum samples were collected in this study, including

43 from VP group and 17 from VD group. We performed 16S rRNA sequencing and metagenomic sequencing on all sputum samples separately. The analysis of microbial alpha and beta diversity in this manuscript is based on 16S, while differential microbial analysis and functional analysis are based on metagenomics. Please see lines 161 and 193 in the revised manuscript.

4) In Fig. 3, it is not clear whether the reported correlations between microbes and genes are positive or negative. Do the microbes promote or suppress the expression of these genes? What are the potential biological mechanisms underlying these associations? At present, the results appear to report raw correlation p-values without correction for multiple testing (q-values), making it likely that many of these correlations arise by chance. In its current form, the findings offer limited insight into the true relationship between the microbiome and the transcriptome.

Answer: Thank you for your valuable suggestion. We used Spearman analysis and FDR correction to recalculate the correlations between microorganisms and between microorganisms and gene expression, and redrawn Figure 3. The cutoff for demonstrating the correlation between microorganisms and genes is that the absolute value of the correlation r-value is greater than or equal to 0.5, and p.adjust is less than 0.05. The line connecting microorganisms and genes represents their relationship. If the line is a solid line, then the two are positively correlated, and if it is a dashed line, then they are negatively correlated. We also add relevant content to the discussion section. Please see lines 235-240 in the revised manuscript.

5) Line 66: The manuscript mentions that "survival rates are reduced", but no clinical outcome data are provided in the results to support this statement.

Answer: Thank you for suggestion. We have revised the inappropriate wording in the abstract section in Lines 64-70.

6) The Introduction section briefly mentions "traditional Chinese medicine", but the manuscript as a whole does not appear to include any analyses or discussion related to Traditional Chinese Medicine. This reference should either be expanded upon or removed for clarity.

Answer: Thanks for suggestion. In the introduction, we supplemented the pathogenesis characteristics of diabetes from the perspective of traditional Chinese medicine, and further discussed in the manuscript Lines 428-441.

7) Sequencing data should be publicly available.

Answer: Thanks for suggestion. We have uploaded the sequencing data to the National Genomics Data Center (NGDC) database and will allow public access on December 1, 2025. Bioproject: subPRO070188. Please see lines 623-627 in the revised manuscript.

8) Line 282: a typo in "16S RNA and birdshot macrogenomic analyses".

Answer: Thanks for suggestion. We have corrected the typo.

Reviewer #2:

This manuscript examines how diabetes influences host transcriptomic responses and respiratory microbiome composition in patients with viral pneumonia, based on integrated analyses of blood and sputum samples from 76 participants. The authors report that diabetes is associated with significant alterations in both host transcriptomic profiles and respiratory microbiome characteristics, including reduced microbial diversity, distinct community composition, and changes in metabolic pathways. Overall, the manuscript is clearly structured, with well-described methods and logically presented results that are clinically relevant. In parts, interpretation could be improved (see comments below).

Answer: Thank you for your suggestion, it is very valuable. We have made revisions to the manuscript.

Some specific comments follow:

1) The sample size is modest, particularly for the transcriptomic analyses (n=27 patients), which limits power to detect subtle differences. Also, the age distribution differs substantially between healthy controls and patient groups, which may confound results. While some comparisons are made between VP and VD, the controls are still considerably younger, complicating the interpretation of "healthy vs. diseased" contrasts. These limitations should be acknowledged more explicitly.

Answer: Thanks for your question. Due to the fact that our patients are from the inpatient population and are influenced by social factors such as an aging population, the admitted population is generally older, resulting in age differences compared to

healthy volunteers. We have elaborated on this limitation in the discussion section.

Please see lines 463-467 in the revised manuscript.

2) The reliance on LefSe with LDA thresholds can sometimes overstate significance when analyzing microbial differences. Have the authors considered using alternative methods (e.g., PERMANOVA) to strengthen the robustness of the α - and β -diversity comparisons?

Answer: Thanks for your question. Alpha diversity and beta diversity reflect the macroscopic differences between two groups. The Shannon index of alpha diversity was calculated using a formula based on sequencing results, and we compared microbial beta diversity using PERMANOVA based on Bray-Curtis. DESeq2 and LefSe are both used to identify differential microorganisms. When conducting LefSe analysis, we screened out microorganisms with LDA scores >2 . Some studies have also adopted LefSe, such as:

[1]. Ye, L., Hou, Y., Hu, W. et al. Repressed *Blautia*-acetate immunological axis underlies breast cancer progression promoted by chronic stress. *Nat Commun* 14, 6160 (2023).

[2]. Zhang, M., Zhang, L., Li, J. et al. Nitrogen-shaped microbiotas with nutrient competition accelerate early-stage residue decomposition in agricultural soils. *Nat Commun* 16, 5793 (2025).

3) The correlation network between transcriptomic and microbial features is interesting, but the statistical thresholds for inclusion (e.g., correlation cutoffs, multiple testing control) are not described in detail. Could the authors provide more information about this?

Answer: Thank you for your valuable suggestion. We used Spearman analysis and FDR correction to recalculate the correlations between microorganisms and between microorganisms and gene expression, and redrawn Figure 3. The cutoff for demonstrating the correlation between microorganisms and genes is that the absolute value of the correlation r-value is greater than or equal to 0.5, and p.adjust is less than 0.05. The line connecting microorganisms and genes represents their relationship. If the line is a solid line, then the two are positively correlated, and if it is a dashed line, then they are negatively correlated. We also add relevant content to the discussion section. Please see Lines 235-240 in the revised manuscript.

4) It is not quite clear if the comorbidities were adjusted for in transcriptome or microbiome analyses, while this is reported in Table 1 (e.g., coronary heart disease was more common in the VD group).

Answer: Thank you for your valuable suggestion. The frequency of VD combined with Coronary Heart Disease (CHD) was higher than that of the VP group. This is consistent with previous research results. As an independent risk factor of coronary heart disease, diabetes patients have a higher prevalence of CHD. We did not correct for this factor in the differential analysis, but we analyzed it in the discussion section

and acknowledged the limitations of this study. Please see lines 385-390 in the revised manuscript.

5) Some conclusions are stated too strongly given the cross-sectional design, which limits causal inference (e.g., statements that diabetes "inhibits" immune regulation or microbial adaptability). Because the analyses rely on inferred functional enrichment from metagenomic data, it would be more accurate to conclude these findings as associations or predicted functional differences rather than definitive mechanistic effects.

Answer: Thank you for your scientific advice. We have revised some of the wording in the manuscript. Expressed as a possible correlation, or the function we predict.

6) The integration of transcriptomic and microbiome findings with traditional Chinese medicine concepts is interesting. However, the link between molecular results and TCM interpretation is sometimes asserted rather than explained. The authors may wish to provide a clearer rationale for how specific gene or microbial changes map onto TCM constructs.

Answer: Thanks for suggestion. In the introduction, we supplemented the pathogenesis characteristics of diabetes from the perspective of traditional Chinese medicine, and further discussed in the manuscript Lines 428-441.

7) Given the large number of differentially expressed genes and microbial taxa, consider providing full results (e.g., DESeq2 output, taxa abundance tables) as supplementary material for transparency and reproducibility.

Answer: Thanks for suggestion. In Supplementary Figure 1, we have included the results of the differential analysis between Deseq2 and LefSe. We have also uploaded the species abundance table in the attachment.

8) Please provide a detailed description of the three figures (and titles), and make sure that all panels are explicitly referenced in the main text (e.g., Figure 1a). It's difficult for readers to follow the connection between results and visuals.

Answer: Thanks for suggestion. We provided a detailed description of the content of the figure in the Figure legend section at the end of the manuscript. And we have added the cited images at the corresponding positions in the manuscript.

9) In Table 1, the entry for "Hospitalizations (Mean {plus minus} SD)" shows a p-value of 9.547. Please verify the calculation and correct the value (or, if 9.547 is a test statistic, report it in footnote separately and provide the corresponding p-value). Also, please specify in the table footnotes the statistical tests used for each variable in Tables 1 and 2.

Answer: Thanks for suggestion. We have verified and modified the p-value of Hospitalizations, and explained the testing methods used for each statistic in Tables 1 and 2, respectively.

10) The formatting of Tables 1 and 2 is not fully consistent (e.g., some variables are reported with two decimals while others are not; some values are center-aligned while others are left-aligned; one table bolds some rows and the other does not; and text fonts appear mismatched). p-value for Height in Table 2 appears to have a typo (double leading 0).

Answer: Thank you for patiently reviewing. We have modified the format of Tables 1 and 2 to ensure consistency in font, alignment, and decimal retention. And the p-value of height was modified.

Re: Spectrum01911-25R1 (Diabetes Affects the Composition of the Respiratory Tract Microbiome and Transcriptome in Patients with Viral Pneumonia)

Dear Dr. Xiaohui Zou:

Thank you for the privilege of reviewing your work. Below you will find my comments, instructions from the Spectrum editorial office, and the reviewer comments.

Revision Guidelines

Sincerely,
Rup Lal
Editor
Microbiology Spectrum

Reviewer #1 (Comments for the Author):

The revised manuscript shows substantial improvement, and most of my initial comments have been addressed appropriately. However, a few minor issues remain that should be corrected:

1. Please specify the statistical tests used before reporting the p-values (e.g., Wilcoxon test, $p < 0.05$; Fisher's test, $p < 0.05$, etc.).

2. It is recommended to include the number of sequencing reads for each sample in the Methods section.
3. Please ensure consistent font formatting throughout the manuscript. For example, the text in lines 241-242 and 369 differs from the rest of the document.

Dear Editor,

On behalf of all the authors, I would like to thank you and all the reviewers for your constructive comments and suggestions on our manuscript. The manuscript has been revised comprehensively according to the editor's and reviewers' comments. We also have the manuscript reviewed. Below are the details of how we have addressed the critiques of all the reviewers/editors.

Reviewer #1 (Comments for the Author):

The revised manuscript shows substantial improvement, and most of my initial comments have been addressed appropriately. However, a few minor issues remain that should be corrected.

Answer: Thank you for your valuable suggestions. We previously overlooked some details, and this revision has been adjusted accordingly.

Some specific comments follow:

1) Please specify the statistical tests used before reporting the p-values (e.g., Wilcoxon test, $p < 0.05$; Fisher's test, $p < 0.05$, etc.).

Answer: Thanks. We have indicated the statistical method used before each P value in the report. Please see lines 255-267 and 322-345 in the revised manuscript.

2) It is recommended to include the number of sequencing reads for each sample in the Methods section.

Answer: Thanks to the reviewer for this valuable suggestion. As recommended, we

have now included the sequencing information for each sample in the Methods section (Lines 180-181, 209-210, and 223-225). Specifically:

For metagenomic sequencing, an average of ~10 Gb of raw data was generated per sample.

For 16S rRNA gene sequencing, an average of ~60,000 raw reads was obtained per sample.

For transcriptomic sequencing, an average of ~6 Gb of raw data was generated per sample.

3) Please ensure consistent font formatting throughout the manuscript. For example, the text in lines 241-242 and 369 differs from the rest of the document.

Answer: Thank you for your patient review. I have rechecked the formatting and ensured consistency throughout.

Re: Spectrum01911-25R2 (**Diabetes Affects the Composition of the Respiratory Tract Microbiome and Transcriptome in Patients with Viral Pneumonia**)

Dear Dr. Xiaohui Zou:

Your manuscript has been accepted, and I am forwarding it to the ASM production staff for publication. Your paper will first be checked to make sure all elements meet the technical requirements. ASM staff will contact you if anything needs to be revised before copyediting and production can begin. Otherwise, you will be notified when your proofs are ready to be viewed.

Sincerely,
Rup Lal
Editor
Microbiology Spectrum

Reviewer #1 (Comments for the Author):

I am satisfied with the given answers and the adjustments in the text. I have no further comments.